

# On the quark spectral function in QCD

Jan Horak[1], Jan M. Pawlowski[1,2] and Nicolas Wink[3]

**1** Institut für Theoretische Physik, Universität Heidelberg,
Philosophenweg 16, 69120 Heidelberg, Germany
**2** ExtreMe Matter Institute EMMI, GSI, Planckstr. 1, 64291 Darmstadt, Germany
**3** Institut für Kernphysik, Technische Universität Darmstadt,
Schlossgartenstraße 2, 64289 Darmstadt, Germany

## Abstract

We calculate the spectral function of light quark flavours in 2+1 flavour vacuum QCD in the isospin-symmetric approximation. We employ spectral Dyson-Schwinger equations and compute the non-perturbative quark propagator directly in real-time, using recent spectral reconstruction results from Gaussian process regression of gluon propagator data in 2+1 flavour lattice QCD. Our results feature a pole-like peak structure at time-like momenta larger than the propagator's gapping scale as well as a negative scattering continuum, which we exploit assuming an analytic pole-tail split during the iterative solution. The computation is augmented with a general discussion of the impact of the quark-gluon vertex and the gluon propagator on the analytic structure of the quark propagator. In particular, we investigate under which conditions the quark propagator shows unphysical complex poles. Our results offer a wide range of applications, encompassing the ab-initio calculation of transport as well as resonance properties in QCD.



# 1 Introduction

The determination of resonance as well as transport properties of QCD requires the knowledge of its fundamental correlation functions for time-like momenta. First principle approaches to QCD are usually formulated in Euclidean spacetime. While the Euclidean formulation is inherent to lattice field theory, in functional approaches such as the functional renormalisation group (fRG) or Dyson-Schwinger equations (DSEs) it is a mere choice as it yields a significant reduction of computational costs. Evidently, within these Euclidean approaches the time-like Minkowski regime is not directly accessible.

An efficient technique of inferring time-like information from Euclidean correlation functions is numerical reconstruction. Due to the ill-conditioned nature of the reconstruction problem, the systematic error of the respective results is an intricate problem, as is the investigation of analytic structure without prior information. Indeed, the two problems are inherently linked: A good grip or even full access to the details of the complex structure facilitate the reconstruction at hand and significantly reduces the systematic error.

A complementary and more attractive approach are direct real-time computations within functional real-time approaches, that can be set-up at the mere expense of significantly increased computational costs. The latter practical problem has to be contrasted with the conceptual ones encountered in real-time formulations of quantum field theories on the lattice. In real-time functional methods though, the central challenge is to cast the equations into a form which allows for an efficient evaluation.

Recently, with the *spectral functional approach*, such a direct real-time has been put forward in [1], including a spectral renormalisation. This approach utilises the fact that spectral representations for correlation functions enable the analytic solution of the loop momentum integrals, and hence the analytic evaluation of the respective functional equation in the entire complex frequency plane. The spectral functional approach has been developed in the framework of Dyson-Schwinger equations (DSEs) and was also applied to Yang-Mills theory

in [2,3]. Its extension to the functional renormalisation group (fRG) has been put forward in [4], applications to scalar theories and gravity can be found in [5,6]. Other direct real-time and complex plane approaches have been put forward, e.g., in DSEs in [7–10], in the fRG in [11–18] and in Bethe-Salpeter equations (BSEs) in [19–24] or on general grounds in [25,26]. For QCD-related reviews on DSEs and the fRG see [27–35] and [36–42].

In this work, we apply the spectral DSE to the quark gap equation and present a direct calculation of the spectral function of light quark flavours in 2+1 flavour vacuum QCD. The computation employs an isospin-symmetric approximation, a bare quark-gluon vertex and assumes an analytic split of the quark spectral function into resonance and scattering contributions. For the gluon propagator we use Gaussian Process Regression (GPR) reconstruction results for the gluon spectral function of 2+1 flavour lattice QCD data [43,44]. In addition to the numerical results, we present a general discussion of the impact of the quark-gluon vertex and the gluon propagator on the analytic structure of the quark propagator. Based on this discussion, we formulate conditions under which complex conjugate poles are present or absent in the quark propagator, and discuss their sources. Reconstructions of the quark spectral function from Euclidean lattice and DSE data in QCD have been previously put forward in [45] respectively [46–48]. Investigations of the quark propagator on the real axis and in the complex plane with various vertex models, also in the context of solving BSEs in the timelike domain, have been put forward in, e.g., [49–66].

Real-time results for the quark propagator have a wide range of possible direct applications: In the case of heavy quarks, the propagator can be directly used to calculate the heavy quark diffusion coefficient, which is a necessary ab-initio input in hydrodynamical simulations of the quark gluon plasma. In the calculation of the QCD resonance spectrum from functional methods the quark propagator is needed for general complex momenta. Using the Källén-Lehmann representation, the quark propagator is readily evaluated in the full complex momentum plane solely using information on the real-time axis.

Our work is organised as follows. A general discussion of the analytic structure of the quark propagator can be found in Section 2. In Section 3, we introduce the spectral quark propagator DSE. Results for the quark spectral function and a discussion of the complex structure of the propagator are presented in Section 4. We conclude in Section 5.

## 2 Analytic properties of the quark propagator

The quark propagator can be parametrised as

$$G_q(p) = \frac{1}{Z_q(p)} \frac{-i\slashed{p} + M_q(p)}{p^2 + M_q(p)^2} \,, \tag{1}$$

with dressing $1/Z_q$ and mass function $M_q$. The singularity structure of $G_q$ is encoded in its universal part

$$g(p) = \frac{1}{p^2 + M_q(p)^2} \,. \tag{2}$$

The singularities of the quark propagator appear as roots of the denominator of (2). The dressing function $1/Z_q$ can be assumed to be singularity-free in the complex plane.

By its gapped nature, the universal part (2) of the quark propagator has to have one or more poles close to the gapping scale $M_q(0)$. These poles can be located either on the first or second Riemann sheet or on their boundary, which is the real axis. On the boundary and on the first sheet, the poles directly show up in the quark propagator as real respectively complex conjugate poles. On the second Riemann sheet, the poles do not directly show up in propagator, but only leave an imprint on the real axis. In the latter case as well as when the

pole is on the real axis, the quark propagator obeys a Källén-Lehmann (KL) representation. In the vacuum, it can then be described by a single scalar function $\rho_q$ via

$$G_q(p) = \int_{-\infty}^{\infty} \frac{d\lambda}{2\pi} \frac{\rho_q(\lambda)}{i\slashed{p} - \lambda}. \tag{3}$$

We drop the spatial momentum argument in the subsequent calculation, as it can be restored from the $\vec{p} = 0$ case via a Lorentz boost due to Lorentz invariance.

The quark spectral function $\rho_q$ can be decomposed into a symmetric and antisymmetric component,

$$\rho_q(\omega) = \rho_q^{(d)}(\omega) + \rho_q^{(s)}(\omega), \tag{4}$$

with

$$\rho_q^{(d)}(-\omega) = \rho_q^{(d)}(\omega), \qquad \rho_q^{(s)}(-\omega) = -\rho_q^{(s)}(\omega). \tag{5}$$

$\rho_q^{(d)}$ and $\rho_q^{(s)}$ account for the Dirac respectively scalar part of the propagator,

$$G_q(p) = -i\slashed{p} \int_\lambda \frac{\rho_q^{(d)}(\lambda)}{p^2 + \lambda^2} + \int_\lambda \frac{\lambda \rho_q^{(s)}(\lambda)}{p^2 + \lambda^2}, \tag{6}$$

with $\int_\lambda = \int_0^\infty d\lambda/\pi$. The components of the spectral function can be obtained separately from the propagator via

$$\begin{aligned}
\rho_q^{(d)}(\omega) &= \frac{1}{2} \mathrm{tr}\Big[\gamma_0 \, \mathrm{Im}\, G_q(-i\omega_+)\Big], \\
\rho_q^{(s)}(\omega) &= \frac{1}{2} \mathrm{tr}\Big[\mathrm{Im}\, G_q(-i\omega_+)\Big],
\end{aligned} \tag{7}$$

where $\omega_+$ denotes the retarded limit, $\omega_+ = \omega + i0^+$. An isolated pole on the real axis shows up as a single Dirac delta contribution in the spectral function. These contributions are associated with stable asymptotic vacuum states, located at the pole mass of the corresponding particle. Note that in gauge theories, these asymptotic states do not necessarily correspond to physically measurable particles [67]. Continuous contributions to the spectral function usually encode the scattering spectrum of the theory. In vacuum, they come with a sharp onset at the energy of the lowest lying scattering state. The quark propagators analytic structure has been intensively studied in the literature [50–52, 63, 66, 68].

The spectral representation (3) also entails the sum rules

$$\int_\lambda \rho_q^{(d)}(\lambda) = \frac{1}{Z_q(p \to \infty)}, \qquad \int_\lambda \lambda \rho_q^{(s)}(\lambda) = 0. \tag{8}$$

Equation (8) can be derived in analogy to Appendix A in [2].

## 2.1 Singularity structure

In the case of poles on the first sheet, the spectral representation (7) is violated. Below, we summarise the discussion of the analytic structure of the quark propagator presented in Appendix A. This discussion eventually motivates the use of the spectral representation of the propagator independent of its precise analytic structure.

Under a small set of well-justified assumptions, which Riemann sheet the poles of the quark propagator appear on is directly linked to the imaginary part of its mass function on the real axis. We assume it to be a smooth and well-behaved function, which is monotonously decaying

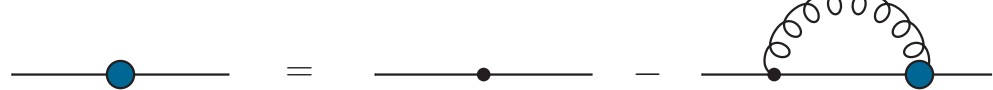

Figure 1: Quark propagator Dyson-Schwinger equation. Notation as defined in Figure 2.

for large complex momenta. Note that this usually holds true due to the logarithmic nature of the branch cut of the propagator. In summary, we find:

*The quark propagator shows a pair of complex conjugate poles on the first Riemann sheet, if*

$$\operatorname{Im} M_q(\omega_0) > 0, \qquad \text{for} \qquad \omega_0^2 - \operatorname{Re} M_q(\omega_0)^2 = 0, \tag{9}$$

where the second part of the equation represents the implicit definition of $\omega_0 \in \mathbb{R}$. An explicit example demonstrating the validity of (9) is given by $M_q(\omega_0) = \omega_0 \pm i\varepsilon$ with $\varepsilon > 0$. With (2), this leads to complex poles on the first/second Riemann sheet for the $+/-$ case, as the sign of the imaginary part of $M$ is flipped going from the upper to the lower half-plane due to the propagator's mirror symmetry. An in-depth discussion of this example is contained in Appendix A.

We emphasise that the above condition for complex poles in the quark propagator is an empirical observation based on a generic mechanism which applies under fairly general conditions in fermionic DSEs. Turning observations such as (9) into rigorous statements is a notoriously hard task in functional methods since analytic structures of correlation functions are highly truncation-dependent. We discuss this matter for the quark gap equation in Section 4.2.2. Details as well as a heuristic derivation of (9) are presented in Appendix A.

In (9), $\omega_0$ is the root of the real part of the universal part $g$'s (2) denominator. It therefore gives the position of the (quasi-) pole of the quark propagator on the real axis. If (9) is fulfilled, the denominator of $g$ has a root in the upper (and lower) right half of the complex plane. If $\operatorname{Im} M_q(\omega_0) = 0$, the quark propagator has a real pole. For $\operatorname{Im} M_q(\omega_0) < 0$, the denominator of (2) most likely will not have a root, and the complex poles move to the second Riemann sheet.

In one-loop perturbation theory, (9) holds true, and the quark propagator shows complex poles. There, as well as in other practical calculations, the imaginary part of the mass function is usually very small in a neighbourhood of $\omega_0$, $\operatorname{Im} M_q(\omega_0) \ll 1$. This entails that independent of which sheet the complex poles lie on, the (quasi-)pole in the universal part (2) can be well approximated by a real pole at $\omega_0$. On the level of the spectral function, this approximation translates into

$$\rho(\omega) = R\,\delta(\omega - \omega_0) + \tilde{\rho}(\omega), \tag{10}$$

for both vector and scalar component, where $R$ is a residue. We call (10) the *resonance-scattering split*.

Equation (10) represents a central approximation of this work. It is used in obtaining all numerical results presented in Section 4. For a detailed discussion of this approximation we refer to Appendix A. There, we also provide the relations between the respective residues and scattering tails of scalar and vector component of the quark spectral function in the resonance-scattering split (10) and the full Minkowski space quark propagator.

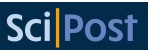

$$S^{(n)} \; = \qquad , \quad \Gamma^{(n)} \; = \qquad , \quad G \; = \qquad$$

Figure 2: Diagrammatic notation used throughout this work: Lines stand for full propagators, small black dots stand for classical vertices, and larger blue dots stand for full vertices. Note that in contrast to other diagrammatic notations, in the present notation, full propagators do not carry blobs, and the classical inverse propagator is simply given by a dot with two legs.

## 3 Spectral quark DSE

In this section, we briefly introduce the spectral quark propagator DSE. This equation is used in obtaining all numerical and analytical results in this work.

The quark propagator DSE can be parametrised as

$$\Gamma_{\bar{q}q}^{(2)}(p) = iZ_2 \, p\!\!\!/ + Z_{m_q} m_q - \Sigma_{\bar{q}q}(p), \tag{11}$$

with the quark self-energy $\Sigma_{\bar{q}q}(p)$ and the wave function renormalisation $Z_2$ and mass renormalisation $Z_{m_q}$ of the quark. In (11) we have used the notation

$$\Gamma_{\phi_1\cdots\phi_n}^{(n)}(p_1,...,p_n) = \frac{\delta^n \Gamma[\phi]}{\delta\phi_1(p_1)....\phi_n(p_n)}, \tag{12}$$

for one-particle irreducible (1PI) correlation functions, derived from the 1PI effective action. In QCD the field $\phi$ encompasses quark, ghosts and gluons, $\phi = (q,\bar{q},A,c,\bar{c})$. The quark-gluon vertex is proportional to $(t^a)^{AB}$ with the gluon color index $a$ and quark and anti-quark indices $A,B$. We can write schematically

$$[\Gamma_{\bar{q}qA}^{(3)}]^a(q,p) = t^a \Gamma_\mu(q,p), \tag{13}$$

where we suppressed the color indices of the quark. The quark self-energy $\Sigma_{\bar{q}q}$ in (11) has the general form

$$\Sigma_{\bar{q}q}(p) = -ig_s C_f \delta^{ab} Z_1^f \gamma_\mu \int_q G_A^{\mu\nu}(q+p) G_q(q) \Gamma_\nu(q,-p), \tag{14}$$

with the strong coupling constant $g_s$ and $Z_1^f$ the quark-gluon vertex renormalisation constant. The combination of (11) and (14) is depicted in Figure 1 with the diagrammatic rules Figure 2.

In (14), the color contractions have already been carried out, yielding the Casimir operator in the fundamental representation, $C_f = 4/3$ for $SU(3)$. $G_A$ and $G_q$ are the gluon respectively quark propagator. The following computation is carried out in the Landau gauge.

We employ the KL representation for the quark and gluon propagators inside the quark self-energy. Then, we apply the scheme of spectral renormalisation [1]. Within this scheme, the momentum integrals in the quark DSE can be solved analytically via standard dimensional regularisation. As a consequence, the quark gap equation can be evaluated *analytically* in the full complex momentum plane, and in particular on the time-like axis. The remaining spectral integrals also require regularisation, which is provided by the spectral renormalisation scheme. The finite spectral integrals are straightforwardly evaluated numerically, and the quark spectral function can be obtained by simply using (7).

Due to its genuine real-time nature, spectral renormalisation permits specifying renormalisation conditions in the Euclidean or Minkowski domain. In particular for massive theories, this allows for on-shell renormalisation. Since the position of the mass (quasi-)pole of the quark is unknown, we refrain from doing so however, and fix the propagator at a large perturbative scale $\mu$ instead,

$$Z_q(p^2 = \mu^2) = 1, \qquad M_q(p^2 = \mu^2) = m_q,$$ (15)

where $m_q$ is the current quark mass. For the explicit calculation we refer to Appendix B.

## 3.1 Gluon propagator

In Landau gauge, the gluon propagator is fully transverse,

$$G_A^{\mu\nu}(p) = \Pi^{\mu\nu}(p)G_A(p), \qquad \Pi^{\mu\nu}(p) = \delta^{\mu\nu} - \frac{p^\mu p^\nu}{p^2},$$ (16)

where $\Pi^{\mu\nu}$ is the transverse projection operator. We employ a spectral representation for the scalar part $G_A(p)$ of the gluon propagator, which reads

$$G_A(p) = \int_\lambda \frac{\lambda \rho_A(\lambda)}{p^2 + \lambda^2}.$$ (17)

Equation (17) implies that all non-analyticities of the gluon propagator are confined to the real momentum axis. In particular, this entails that the gluon propagator does not have complex poles. In fact, the complex structure of the gluon propagator is subject of an ongoing debate. Equation (17) is therefore to an assumption. We discuss implications as well as deviations of it by, e.g., complex conjugate poles in Section 4.2.2.

The gluon spectral function $\rho_A$ represents a non-trivial input for our calculation. We use reconstruction results of 2+1 flavour lattice QCD data [43,44] obtained via Gaussian process regression in [69] for $\rho_A$, shown in Figure 3. The lattice data corresponds to a decoupling-type solution, i.e., the gluon propagator approaches a finite, non-zero value in the origin, as shown in the inset of Figure 3.

## 3.2 Quark-gluon vertex

The existence of analytic solutions for the momentum loop integrals is at the heart of spectral functional approaches. This imposes restrictions on the representations of the correlation functions entering. For example, full vertices or particular momentum channels thereof can be included via their spectral representations, as done, e.g., in [1].

Due to the transversality of the gluon propagator in the Landau gauge, only the transverse part $\mathbf{\Gamma}_\mu(p,q)$ of the quark-gluon vertex $\Gamma_\mu(p,q)$ enters the gap equation. We define

$$\mathbf{\Gamma}_\mu(p,q) = \Pi_{\mu\nu}(p+q)\Gamma_\nu(q,p),$$ (18)

with the transverse projection operator (16). While the full quark-gluon vertex can be expanded in a basis with twelve tensor components, the transverse vertex $\mathbf{\Gamma}_\nu$ is expanded in eight transverse tensor structures,

$$\mathbf{\Gamma}_\mu(p,q) = g_s \sum_{i=1}^{8} \lambda_i(p,q)\mathcal{T}_i(p,q).$$ (19)

Here, $q, p$ are the (incoming) anti-quark and quark momenta respectively and the incoming gluon momentum is $-(p+q)$, see, e.g., [70].

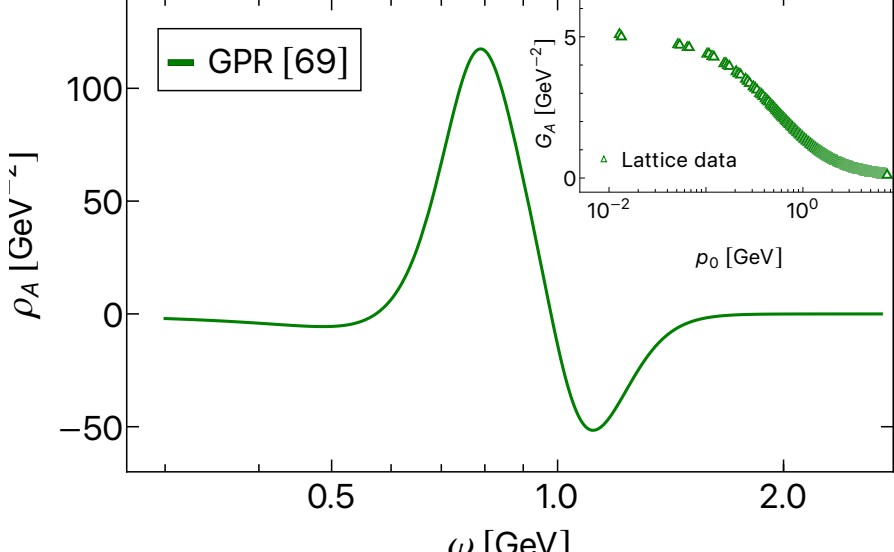

Figure 3: Gluon spectral function obtained via Gaussian process regression in [69] from 2+1 flavor lattice QCD data [43,44]. The spectral function shows a characteristic gapping scale at about $\omega_A \approx 800$ MeV. IR and UV asymptotics are negative, as required analytically, for details see [69]. The lattice data is shown in the inset on the top right and corresponds to a decoupling-type gluon propagator.

While the tensor basis $\{\mathcal{T}_i\}$ is not unique, it can be ordered such that it only hosts three dominant components [70–74]. Naturally, one of them is the chirally symmetric classical tensor structure, and we choose $\mathcal{T}_1 = \gamma_\mu$, or rather its transverse projection,

$$\mathcal{T}_1(q,p) = -\mathrm{i}\,\gamma_\mu\,\Pi_{\mu\nu}(p+q)\,. \tag{20}$$

The choice for the dominant components is completed by a further chirally symmetric tensor structure and one, that breaks chiral symmetry. The dressing $\lambda_1$ of the classical tensor structure can be related to the wave function renormalisation of the quark as well as a scattering kernel via the Slavnov-Taylor identities (STIs), see, e.g., the reviews [28,30] for more details. The representation of the STIs and their impact in the present work relies on [70,74], and we refer to these works for further discussions.

Roughly speaking, the STIs for the quark-gluon vertex express the four longitudinal dressings in terms of the quark propagator and additional scattering kernels, that carry the quantum modifications of BRST transformations. While the general relations are rather complicated, they are significantly reduced under the assumption that the quantum BRST transformations are approximated well by the classical ones. Indeed, in general the scattering kernels show a rather mild momentum dependence [72,74,75], supporting this approximation. An exception is triggered by the Schwinger mechanism for confinement that necessarily leads to longitudinal poles in the ghost-gluon scattering kernels. Without the scattering kernels, the STI for the quark-gluon vertex takes the simple Abelian form

$$(p_\mu + q_\mu)\Gamma_\mu(q,p) = g_s \left[ \Gamma^{(2)}_{\bar{q}q}(q) - \Gamma^{(2)}_{\bar{q}q}(p) \right], \tag{21}$$

identical to the form of the Ward identity in QED.

The approximation of the quantum BRST transformations as classical ones discussed above are at the heart of the Ball-Chiu construction [76,77] and variants thereof, e.g., [78–80], for a discussion see [28]. All these vertices are constructed around a unique combination of the quark dressings $Z_q$ and $M_q$. The difference between the variants of this construction is an

undetermined additional transverse part dropping out of the STIs. Effectively, STI vertices rely on the smallness of this additional piece. The construction is Abelian and also works in $U(1)$ theories such as QED.

Consistent approximations of the quark-gluon vertex, leading to the right amount of chiral symmetry breaking, are intricate: To begin with, as discussed before, quantitatively reliable results are only obtained self-consistently from the coupled set of propagator and quark-gluon vertex DSEs, if we consider at least three out of the eight transverse tensor structures.

In gauge theories we face the situation that the correct complex structure of propagators and vertices may only be obtained in a fully gauge-consistent approximation, for a discussion see, e.g., [3]. This suggests an investigation of the quark gap equation with STI quark-gluon vertices. However, while the dressing of the classical tensor structure is constrained by the STIs, the dressings of the other two relevant tensor structures are not. Hence, one may drop them in a first attempt on the complex structure of the quark. In this case, the physical amount of chiral symmetry breaking can only be obtained by an infrared enhancement of the dressing $\lambda_1$. This may introduce an additional complex structure into the gap equation, whose impact is hard to control. A similar analysis has been done very thoroughly in the pure Yang-Mills system, see [81]: The ensuing location and strength of complex singularities of ghost and gluon varied greatly, depending on the vertex dressings. This suggests that a conclusive study involves a self-consistent computation using all the three dominant tensor structures and an STI-consistent dressing for the classical tensor structure. We discuss the complex structure of the quark propagator in the scenario with STI vertices in Section 4.2.2. A quantitative computational analysis of this scenario goes beyond the scope of the present work and will be presented elsewhere.

In the present work we close the remaining gap in approximations studied in the literature: We numerically solve the gap equation with the input of a full real-time gluon propagator augmented with classical vertices,

$$\Gamma_\nu \approx -i g_s \gamma_\nu. \tag{22}$$

In this approximation, the gap equation takes the simple form

$$\Sigma_{\bar{q}q}(p) = -g_s^2 C_f \delta^{ab} Z_1^f \int_\lambda \int_q G_A^{\mu\nu}(q+p) \gamma_\mu \frac{1}{\slashed{q} - \lambda} \rho_q(\lambda) \gamma_\nu. \tag{23}$$

To account for a sufficient amount of chiral symmetry breaking, we amplify the value of the coupling constant such that the correct constituent quark masses are produced. Note that such a qualitative procedure necessarily leads to quark propagators with an enhanced ultraviolet tail. We emphasise that in this work, we do not aim at quantitatively improving its description in the Euclidean domain. Instead, the purpose of this study is a non-perturbative, direct investigation of the complex structure of the quark propagator using a full 2+1 flavour QCD gluon propagator.

## 4 Results

The spectral quark DSE introduced in Section 3 is solved in 2+1 flavour QCD for light quark flavours using the isospin-symmetric approximation. For the gluon propagator, we use the input discussed in Section 3.1. Our quark-gluon vertex truncation, the classical vertex approximation, is discussed in detail in Section 3.2. The strong coupling constant is set to $\alpha_s = 1.11$ such that in the chiral limit, a dynamical constituent quark mass of $M_\chi(0) \approx 350$ MeV is generated. Formally, the current quark mass for light flavours is fixed through the pion mass $m_\pi$ and decay constant $f_\pi$ by the Gell-Mann-Oakes-Renner (GMOR) relation. Here, we treat $m_q$

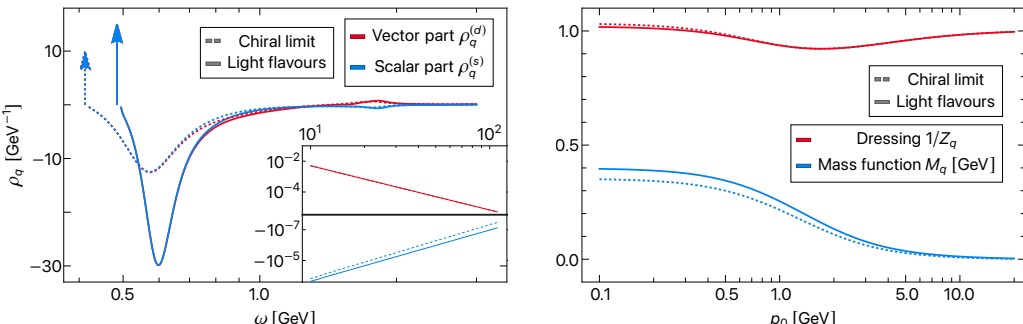

Figure 4: DSE results for the quark propagator in the chiral limit (dashed) and for light quark flavours (solid). Left: Spectral function in the resonance-scattering split (10). Arrows indicate delta poles. Their height encodes the relative size of the residues. The numerical values of the pole positions and residues are given in Table 1. Note that the appearance of the delta poles is due to the resonance-scattering split. The scattering tail is predominantly negative, as necessitated by the sum rules (8). While the large negative peak in the scattering tail can be understood as part of the pole structure, the bump at about 1.8 GeV is connected to quark-gluon scattering. Right: Euclidean dressing and mass function. In the chiral limit, a constituent quark mass of $M_\chi(0) = 353$ MeV is obtained. For light quark flavours in the isospin symmetric approximation, we find $M_l(0) = 398$ MeV. The dressing function approaches a value above one in the IR. This feature is attributed the use of a classical quark-gluon vertex in Landau gauge, and explicitly investigated in Appendix C.

as a phenomenological parameter instead. For $m_q = 1.2$ MeV, we obtain a constituent quark mass for light quark flavours of $M_l(0) \approx 400$ MeV.

The renormalisation conditions (15) are employed at $\mu = 30$ GeV. All solutions are obtained using the resonance-scattering split (10).

## 4.1 Numerical results

Our results for the quark spectral function for light quark flavours and in the chiral limit are shown in the left panel of Figure 4. Since the resonance-scattering split was employed, all spectral functions feature a genuine delta pole, with positive residue. The pole position is identical for both components, since it follows from the universal part of the quark propagator (2).

The peak moves towards larger frequencies going from the chiral limit towards the light flavours, as expected from the increase of the constituent quark mass. The residues of both components increase accordingly, since the mass function respectively the peak position appear in the residues, cf. (A.15). The residues of both components are nearly identical, as anticipated from (A.16). The numerical values of the peak positions and residues can be found in Table 1. We note that while the constituent quark mass increases about 50 MeV from the chiral limit to light flavours, the pole position moves up about 70 MeV. Since also the scattering tail grows in amplitude, a larger increase in the pole mass than in the constituent mass is necessary. The scattering tail is negative and therefore yields a negative contribution to the value of the mass function in the origin.

The scattering tails of both, vector and scalar components are predominantly negative. Since the residues are positive, this is necessitated by the quark spectral functions normalisation condition (8). Remarkably, for both components we find the sum rules to be fulfilled with an accuracy of about 1 ‰. We interpret the negative bump directly to the right of the delta pole

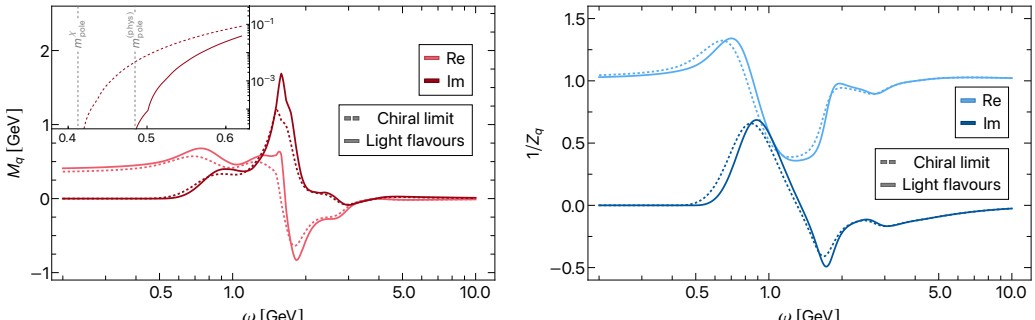

Figure 5: Results for the quark propagator in Minkowski space in the chiral limit (dashed) and for light quark flavours (solid). Left: Mass function. The imaginary part shows a sharp onset at the respective pole mass $m_{\text{pole}}$, as shown in the inset. At about 1.8 GeV, the imaginary part shows a peak, while the real part has a zero crossing. These structures are attributed to quark-gluon scattering, compare the corresponding bump in the spectral function in the left panel of Figure 4. Right: Dressing function. The imaginary part shows a (negative) peak at about the quark-gluon scattering scale as well.

as part of the actual pole structure, part of which is approximated by a delta pole in the resonance-scattering split. We discuss this in more detail in Section 2.1.

The second particular structure in the scattering tail is a small bump at about 1.8 GeV, which has opposite sign for vector and scalar components. We relate this structure to quark-gluon scattering: For the light flavours, its position is approximately at twice the quark mass pole position $\omega_0$ plus the peak position of the gluon spectral function $\omega_A$, i.e., $\omega_{\text{bump}} \approx 2\omega_0 + \omega_A$. Hence, the structure can be understood as a washed-out onset for quark-gluon scattering. This interpretation is supported by the fact that, when using a more strongly peaked gluon spectral function, e.g., the Yang-Mills reconstructions results of [82], the structure becomes much sharper and more pronounced. In the chiral limit, the bump position is less pronounced and difficult to locate, appears at a similar scale, however.

The inset in the left panel of Figure 4 shows the ultraviolet tails of both spectral function components. The opposite sign follows from the different anomalous dimensions of the quark wave and mass functions.

In the right panel of Figure 4 we display the Euclidean quark mass and dressing function. In the chiral limit, we obtain $M_\chi(0) = 353$ MeV, whereas for the light flavours, we have $M_l(0) = 398$ MeV. The dressing function $1/Z_q(p)$ rises above one in the deep IR, in contrast to results using more sophisticated vertex truncations [70, 72, 74, 83, 84]. We therefore attribute this feature to our vertex truncation, which can be considered too crude to yield quantitative statements about the Euclidean domain. In Appendix C we demonstrate that at one-loop level, the IR behaviour of the dressing function is strongly gauge parameter dependent.

Table 1: Results for the numerical parameters in the resonance-scattering split (10) describing the pole contribution of the quark spectral functions in Figure 4. We numerically verify that $R^{(s)} \approx R^{(d)}$, according to (A.16).

|  | Chiral limit | Light flavours |
|---|---|---|
| $m_{\text{pole}}$ [GeV] | 0.412 | 0.485 |
| $R^{(s)}$ | 0.768 | 1.15 |
| $R^{(d)}$ | 0.766 | 1.15 |

## 4.2 Analytic structure

Next, we present a general discussion of the analytic structure of the quark propagator in light of the numerical findings presented in the previous section. We start by discussing the effects of our truncation with classical vertices and the resonance-scattering split on the analytic structure of the result in Section 4.2.1. In Section 4.2.2, in particular the impact of STI vertex constructions and non-spectral gluon propagators are discussed.

### 4.2.1 Present truncation

Our truncation of the spectral quark DSE, discussed in detail in Section 3, along with resonance-scattering split (10) allows us to discuss the analytic structure of the quark propagator within the following scenario:

(i) The gluon propagator obeys a Källén-Lehmann representation and does not exhibit poles,

(ii) the quark propagator has a pole on the real axis at $\omega_0 = m_{\text{pole}}$,

(iii) only the classical tensor structure of the quark-gluon vertex contributes and its dressing is constant.

Note that certainly, (iii) does not hold true for the full quark-gluon vertex. The latter can be expected to show an imaginary part as well as a branch cut for specific combinations of real frequencies. As vertex models with (iii) are still used in practical calculations, we include this situation in the present discussion.

Under the above conditions, the universal part of the propagator (2) features a branch cut with support in $[m_{\text{pole}}, \infty)$. Since the branch cut of the gluon propagator starts in the origin, the branch point $m_{\text{pole}} \in \mathbb{R}$ of the quark propagator arises at the same point as its pole. This can be easily seen from the scalar part of the quark self-energy diagram, see (B.7). In consequence, the universal part shows a *branch point singularity*. Note that this alone does not necessarily violate the KL representation.

In addition to the branch point singularity, the propagator can show complex poles. As argued in Section 2, the presence or absence of complex poles depends on the sign of the mass functions imaginary part in the neighbourhood of the branch point $m_{\text{pole}}$. If the imaginary part is negative, the poles are on the second Riemann sheet and the quark propagator obeys the KL representation. If it is positive, the pole are on the first sheet and the KL representation is violated, see (9). In the left panel of Figure 5 we display the mass function on the real axis. We find that the imaginary part of the mass function approaches zero from above at $m_{\text{pole}}$. Although strictly speaking, (9) does not apply here since Im $M_q(m_{\text{pole}}) = 0$, our quark propagator still shows *complex poles*. This can be validated by calculating the corresponding spectral function with (7) without the resonance-scattering split (10): It shows a negative instead of a positive peak around $m_{\text{pole}}$, entailing that the propagators branch point singularity is negative. In consequence, the corresponding Euclidean propagator obtained through the spectral representation (3) does not reproduce the Euclidean results in the right panel of Figure 4.

We emphasise that the real pole of the spectral function is a built-in feature of the resonance-scattering split (10) and does not correspond to the quark propagators analytic structure described above. Rather, our results suggest that the quark propagator can be very well approximated by the spectral functions in this approximation, displayed in Figure 4. In contradistinction, the condition for the appearance of complex poles formulated in (9) is general. An evaluation confirming the applicability of the resonance-scattering split at the level of the Euclidean propagator is presented in Appendix A.1.

### 4.2.2 STI vertices & non-spectral gluon propagators

The full analytic structure of the quark propagator is a long-standing question. A clear answer is hindered by the need to truncate the infinite tower of diagrammatic equations for QCD correlation functions in functional approaches. For example, in [81] a strong dependence of the analytic structure of the gluon propagator on vertex models has been observed via a direct calculation in Yang-Mills theory for complex frequencies. For the quark propagator such a strong dependence has been observed in [52,54,57,85], where complex conjugate poles in the quark propagator where found depending on the vertex model. In the following, we discuss the impact of the two input correlation functions in the quark gap equation on the analytic structure of the quark propagator: The quark-gluon vertex and gluon propagator. The discussion will put focus on sources of complex non-analyticities, which are consistent approximation of the full quark-gluon vertex and the complex structure of the gluon. To that end, we will use the examples of an STI vertex construction and a non-spectral gluon propagator.

**Quark-gluon vertex** As discussed in Section 3.2, the quark-gluon vertex is constrained by an STI which implements gauge consistency of the solution. A spectral vertex model fulfilling the particularly simple Abelian approximation (24) of the STI has been introduced in [86] and reads,

$$G_q(q)\Gamma_\mu(q,p)G_q(p) \approx g_s \int_\lambda \frac{1}{i\slashed{q} - \lambda}\gamma_\mu\frac{1}{i\slashed{p} - \lambda}\rho_q(\lambda). \tag{24}$$

Equation (24) directly builds on the quark spectral representation (3) with the spectral function of the quark propagator $\rho_q$. Contracting (24) with $(p+q)_\mu$ leads to (21), multiplied from the left and right by quark propagators.

The STI vertex (24) can be directly used in the gap equation (14) when multiplying the latter with the quark propagator $G_q(p)$. Then, (14) reduces to

$$1 = \left(iZ_2\slashed{p} + Z_{m_q}m_q\right)G_q(p) + g_s^2 C_f Z_1^f \int_\lambda \int_q G_A^{\mu\nu}(q+p)\gamma_\mu\frac{1}{i\slashed{q} - \lambda}\gamma_\nu\frac{1}{i\slashed{p} - \lambda}\rho_q(\lambda). \tag{25}$$

By taking the imaginary part of (25), we project onto the spectral function of the quark propagator in the first term on the right hands side. Hence, the Abelian STI vertex defined in (24) reduces the gap equation from an initially non-linear to a linear equation for the quark spectral function. This greatly simplifies the task of numerically solving the equation.

In QED, (25) can be solved analytically if augmented with a classical spectral function $\rho_A$ of a photon, i.e., a simple delta pole, as done in [8]. The resulting spectral function of the electron has the pole contribution at the electron mass as well as a scattering tail. Hence, no violation of the KL representation of the electron propagator has been found there. In turn, the electron propagator derived from the gap equation with classical vertices (23) has complex conjugate poles, in analogy to our results, see the discussion in Section 4.2.1. This shows impressively that gauge consistency of correlation functions plays a pivotal role for the existence of spectral representations. This observation can also heuristically be linked to the form of the gap equation with STI vertices: The STI turns the initially non-linear equation for the quark spectral function (11) with a complex solution into a linear one with a real solution (25).

**Gluon propagator** A pivotal ingredient in above discussion is the simple *spectral* structure of the photon/gluon propagator. In contrast to the photon spectral function, the spectral representation of the gluon is far more complicated. An extended representation features a spectral (KL) part and additional complex singularities.

The spectral part necessarily show negative parts at asymptotically large and small spectral values [82] in contradistinction to the positive photon spectral function. Moreover, complex poles have been observed in direct real-time calculations of the gluon propagator in different truncations of Yang-Mills theory [3, 81]. It has been argued in [3] that these poles render a consistent solution difficult and, as stated above, might ultimately be linked to inconsistencies in the respective truncation. Finally, precision reconstructions of Yang-Mills and QCD gluon propagators have been performed in a purely spectral manner without complex poles [69, 82, 87, 88], and with complex poles [89–92].

A gauge boson propagator with complex conjugate poles can lead to further complex conjugate cuts in the quark propagator, violating the KL representation. This has been shown in [3] in the ghost-gluon system. The mechanism is very general: The complex singularities of the gluon propagator are dragged along by the loop momentum integration to form branch cuts in the ghost self-energy. Note that in the same fashion, the ordinary branch cut on the real axis in polarisation diagrams emerges from integrating two massive propagators. Therefore, the mechanism also applies in the quark gap equation with bare vertices, and complex conjugate cuts are produced when using a gluon with complex poles. The analytic structure w.r.t to loop momentum of the integrands in the gap equation with classical vertices (11) and with STI vertex (25) is identical, however. This leads us to the following result about the analytic structure of the quark propagator:

*For a gluon with complex conjugate poles, the quark gap equation with either classical vertices (22) or STI vertices (24) leads to a violation of the spectral representation of the quark by complex conjugate cuts.*

This observation may have far-reaching consequences for the direct real-time computation of scattering elements via DSE, Bethe-Salpeter, Faddeev and four-body equations, which will be discussed elsewhere.

## 5   Conclusion & Outlook

In this work, we computed the quark spectral function of vacuum QCD for light quark flavours assuming isospin symmetry. The full quark propagator was obtained by solving its spectral Dyson-Schwinger equation using 2+1 flavour lattice QCD gluon propagator data [43, 44] and a classical quark-gluon vertex. We employed a spectral representation for the gluon propagator and used reconstruction results from Gaussian process regression [69] for the gluon spectral function.

In this approximation, the quark propagator shows a pair of complex conjugate poles located very close to the real axis. Nevertheless, the quark propagator can be very well approximated to obey a Källén-Lehmann representation by using an analytic split into resonance and scattering contribution for the quark spectral function, cf. Section 2.1. Within this approximation, the quark spectral function shows a delta pole and a continuous scattering contribution. While the delta pole features a positive residue, the scattering tail is predominantly negative. The latter fact is necessitated by the sum rule of the quark spectral function.

In QED, it has been found that complex poles in the electron gap equation disappear when using STI-consistent vertex constructions. Similar observations have been made indirectly in QCD, implying that the quark obeys a spectral representation in this case. Even in the case of STI-consistent vertices this property is lost again when using a gluon propagator with complex conjugate poles also, as we argue in Section 4.2.2.

Our investigation represents a further step towards understanding the time-like structure of fundamental correlation functions of QCD. To further deepen this understanding, the impact of different quark-gluon vertex models on the complex structure of the quark propagator should

be studied. Using an STI-consistent vertex construction together with a full gluon propagator in the spectral quark DSE can be regarded as a natural next step in this direction. Our approach enables to directly resolve the full complex momentum plane for different vertex models and gluon propagators.

Our results for the quark spectral function, including systematic improvements of the current truncation, have a wide range of possible applications. In hydrodynamical simulations of the quark gluon plasma, QCD transport coefficients represent necessary input which can only be calculated from ab initio methods such as functional methods. The transport coefficients are linked to fundamental correlation functions via Kubo relations. Of particular interest is the heavy quark diffusion coefficient. Due to their massive nature, heavy quarks act as probes of the thermodynamic evolution of the quark gluon plasma. Our calculation for light quarks extends straightforwardly to that of the heavy quark propagators in real-time. Indeed, the systematic error of the present approximation is significantly reduced in the latter case, as both the quark propagator and the respective vertex carry less (chiral) dynamics. The calculation of the QCD hadron spectrum represents another promising application. Resonances can be determined from functional methods by solving resonance equations such as the Bethe-Salpeter equation in the time-like domain. As an input, the fundamental QCD correlation functions in the complex momentum plane are required. Here, our results for the quark propagator can directly be used.

# 6 Acknowledgements

We thank G. Eichmann and J. Papavassiliou for discussions. This work is done within the fQCD-collaboration [93], and we thank the members for discussion and collaborations on related projects.

**Funding information** This work is funded by the Deutsche Forschungsgemeinschaft (DFG, German Research Foundation) under Germany's Excellence Strategy EXC 2181/1 - 390900948 (the Heidelberg STRUCTURES Excellence Cluster) and the Collaborative Research Centre SFB 1225 (ISOQUANT). JH acknowledges support by the Studienstiftung des deutschen Volkes. NW acknowledges support by the Deutsche Forschungsgemeinschaft (DFG, German Research Foundation) - Project number 315477589 - TRR 211 and by the State of Hesse within the Research Cluster ELEMENTS (Project ID 500/10.006).

# A Resonance-scattering split

The singularity structure of the propagator is entirely determined by its universal part $g$. For real frequencies, it reads

$$g(\omega_+) = \frac{1}{M_q(\omega_+)^2 - \omega_+^2} \,. \tag{A.1}$$

In (A.1) and in the following, we make use of the notation $M_q(\omega_+) = M_q(p = -i\omega_+)$, with the retarded limit $\omega_+ = \omega + i0^+$. If $G_q$ obeys the KL representation, so does $g$, with a spectral function

$$\rho_g(\omega) = 2 \operatorname{Im} g(\omega_+) \,. \tag{A.2}$$

Since in (A.1) the retarded limit $\omega_+$ is considered, the subsequent discussion applies to the complex upper half plane. The pole(s) of the quark propagator appear as the roots of the

denominator of (A.1). We distinguish three cases: Real, complex and no roots. For a real root,

$$\omega_0 - M_q(\omega_0) = 0\,, \tag{A.3}$$

with $\omega_0 \in \mathbb{R}$, one simply obtains an ordinary massive particle pole. Already at one-loop order in perturbation theory however, the mass function $M_q$ obtains a non-vanishing imaginary part on the positive real axis, such that a real root is no longer possible. We consider this imaginary part to be a small, constant imaginary perturbation in a neighbourhood of the previously real pole $\omega_0$ of the mass function in (A.3), i.e.,

$$M_\varepsilon(\omega_c) = M_q(\omega_c) \pm i\varepsilon\,, \quad \text{for} \quad |\omega_c - \omega_0| \ll 1\,, \tag{A.4}$$

with $\varepsilon \ll 1$ and $\omega_c \in \mathbb{C}$ now. In (A.4), $M_q$ is to be understood as the real part of $M_\varepsilon$, i.e., $M_q(\omega) \in \mathbb{R}$. Then, $M_q$ has no branch cut, we can omit the retarded limit reminding us which side of the branch cut we are on and simply write $M_q(\omega)$ on the real axis. Due to the propagators mirror symmetry, also the mass function obeys

$$M_\varepsilon(\bar\omega_c) = \bar M_\varepsilon(\omega_c)\,. \tag{A.5}$$

With (A.4), the quark propagators complex poles are given by the solution to

$$\omega_c - M_\varepsilon(\omega_c) = 0\,. \tag{A.6}$$

Since we are working in the upper half plane, we have $\text{Im }\omega_c > 0$. Then, as $M_\varepsilon$ obeys (A.5), (A.6) only has a solution for $\text{Im }M_\varepsilon > 0$, i.e. the plus case in (A.4). In this case, the complex poles appear on the first Riemann sheet, and show up in the propagators. For the $\text{Im }M < 0$ case, the complex poles are located on the second Riemann sheet, and hence do not appear in calculations.

For the quark spectral representation, above consideration have the following consequences: For $\text{Im }M_\varepsilon < 0$, the spectral representation is intact. The corresponding universal quark spectral function shows a distinct, sharp positive peak structure around $\omega_0$ plus a scattering continuum. We will focus on the peak structure in the following.

Since the imaginary part of the mass function (A.4) is small, due to the Sokhotski-Plemelj theorem,

$$\lim_{\varepsilon \to 0} \text{Im } \frac{1}{(M_q(\omega) \pm i\varepsilon)^2 - \omega^2} = \pm \frac{\pi}{2\omega_0} \delta(\omega - \omega_0)\,, \tag{A.7}$$

the peak is well approximated by a delta distribution. Note that also in (A.7), we only considered the positive frequency contribution. We thus have

$$\rho_g(\omega) \approx \frac{\pi}{\omega_0} \delta(\omega - \omega_0)\,. \tag{A.8}$$

On the contrary, for $\text{Im }M_\varepsilon > 0$ the KL representation is violated by the complex poles. In this case, the universal part of the quark propagator (A.1) can be described by a modified spectral representation which explicitly takes the complex poles into account. On the real axis, it reads

$$g(\omega_+) = \frac{1}{(\omega_0 + i\varepsilon)^2 - \omega_+^2} + \frac{1}{(\omega_0 - i\varepsilon)^2 - \omega_+^2} + \int_\lambda \frac{\lambda\rho_g(\lambda)}{\lambda^2 - \omega_+^2}\,, \tag{A.9}$$

with $\rho_g$ as defined in (A.2). As for the $\text{Im }M_\varepsilon > 0$ case, with (A.7) we find that the spectral functions $\rho_g$ shows a sharp peak around $\omega_0$ with negative residue

$$\rho_g(\omega) \approx -\frac{\pi}{\omega_0} \delta(\omega - \omega_0)\,. \tag{A.10}$$

In this case, the universal part (A.9) then evaluates to

$$g(\omega_+) \approx \frac{1}{(\omega_0 + i\varepsilon)^2 - \omega_+^2} + \frac{1}{(\omega_0 - i\varepsilon)^2 - \omega_+^2} - \frac{1}{\omega_0^2 - \omega_+^2}. \qquad (A.11)$$

Taking the imaginary part in (A.11), we recover (A.10).

Note that the real pole part in (A.9) enters with a minus sign. Since $\varepsilon \ll 1$ in (A.11), for any complex frequency not in the direct vicinity of $\omega_0$, the real pole effectively cancels on of the complex poles. In other words, the complex pole universal part (A.11) is practically indistinguishable from a single massive propagator, i.e.,

$$g(\omega_c) \approx \frac{1}{\omega_0^2 - \omega_c^2}, \quad \text{for} \quad |\omega_c - \omega_0| \gtrsim \varepsilon, \qquad (A.12)$$

which applies in particular on the Euclidean axis. From the perspective of a generic lower limit on the numerical resolution in the complex plane, the approximation (A.12) is therefore well justified. It says that the sum of the two complex poles with positive residue and the negative real quasi-pole is well approximated by a single real positive pole. In terms of the spectral function, this has the consequence that the spectral representation of the universal part and hence of the quark propagator itself is restored. The spectral function is then simply given by the Im $M_\varepsilon < 0$ case (A.8).

Above considerations suggest that in the case of a small imaginary part in the mass function, independent of its sign and the resulting particular analytic structure, the quark propagator can be well approximated to obey a KL representation. In this case, the universal spectral function is well represented by an analytical split into a genuine pole contribution plus a continuous scattering tale, which is

$$\rho_g(\omega) = \frac{\pi}{\omega_0} \delta(\omega - \omega_0) + \tilde{\rho}_g(\omega), \qquad (A.13)$$

with scattering tail $\tilde{\rho}_g$. In terms of the Dirac and mass components of the spectral function, this pole-tail split reads

$$\rho^{(d/s)}(\omega) = R^{(d/s)} \delta(\omega - \omega_0) + \tilde{\rho}^{(d/s)}(\omega). \qquad (A.14)$$

The scattering contribution $\tilde{\rho}^{(d/s)}$ was neglected in above consideration but does not influence the discussion, as the quasi-pole structure is extremely sharp and distinct. The pole structure dominates the entire IR behaviour of the propagator in the sense of a gapping, while the scattering tail gets relevant towards in the UV and in particular carries the perturbative information. We emphasise that the applicability of this split depends on the imaginary part of the mass function and always has to be tested empirically.

In the pole-tail split (A.14), the residues $R^{(d/s)}$ are given by

$$R^{(d)} = \pi \omega_0 \, \text{Re} \, \frac{1}{Z_q(\omega_0)}, \qquad R^{(s)} = \pi \, \text{Re} \, \frac{M_\varepsilon(\omega_0)}{Z_q(\omega_0)}. \qquad (A.15)$$

Note that since $M_q(\omega_0) = \omega_0 \pm i\varepsilon$, we have

$$R^{(d)} \approx R^{(s)}. \qquad (A.16)$$

The scattering tails in (A.13) are still given by (7), augmented with a suitable lower for the spectral tail such that the pole contribution is not included. This cut-off is related to the width of the pole and hence to the distance of the complex poles to the real axis. The numerical value for the cut-off used in our implementation is specified in Appendix D.3.

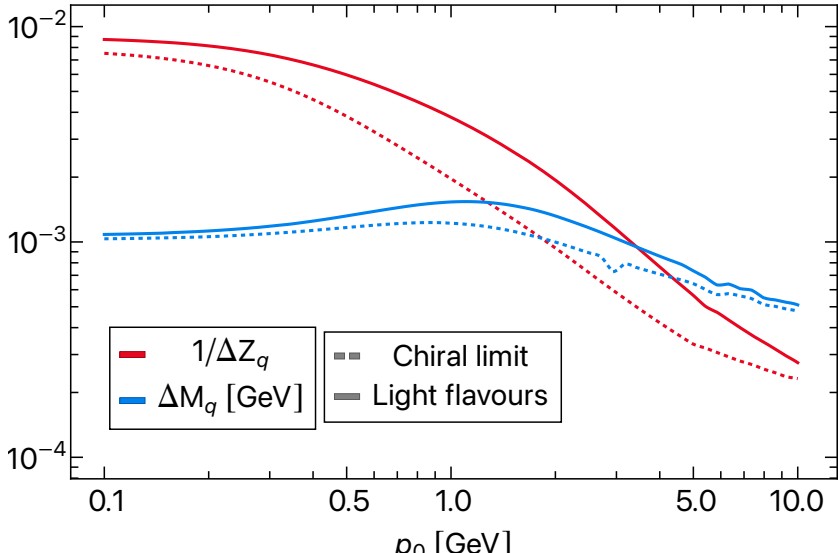

Figure 6: Evaluation of the quality of the resonance-scattering split (10) introduced in Section 2. We consider the Euclidean mass and dressing function and compare their difference in absolute value from the spectral representation to that directly obtained from the spectral Euclidean DSE. The deviation in the mass function does not supersede 10 MeV, while that for the dressing function is about $10^{-3}$. Comparing to the absolute values of mass and dressing function Figure 4, we conclude that the resonance-scattering split represents the quark propagator quantitatively very well.

### A.1 Quality of the approximation

The validity of the resonance-scattering split is tested by comparing the Euclidean mass and dressing function obtained through the spectral representation (3) with the spectral functions shown in Figure 4 against their counterparts obtained directly from the spectral Euclidean DSE. In Figure 6, we show their absolute difference. If the spectral representation is intact, the difference should be zero within the accuracy of our numerical integration. Our numerical integration routine aims at relative accuracy of about $10^{-3}$. We conclude that the approximation works well on a quantitative level. The error from neglecting the imaginary part of the poles is visible in particular in the mass function, but seems to be of negligible size.

## B Self-energy calculation

Equation (6) makes explicit how the components of the spectral function are related to those of the propagator, i.e.,

$$
\int_\lambda \frac{\rho_q^{(d)}(\lambda)}{p^2 + \lambda^2} = \frac{1}{Z_q(p)} \frac{1}{p^2 + M_q(p)^2} \, ,
$$

$$
\int_\lambda \frac{\lambda \rho_q^{(s)}(\lambda)}{p^2 + \lambda^2} = \frac{M_q(p)}{Z_q(p)} \frac{1}{p^2 + M_q(p)^2} \, .
$$

(B.1)

We will use (6) in the following calculation of the spectral quark self-energy diagram. Using a bare quark-gluon vertex, the diagram reads

$$\Sigma_{\bar{q}q}(p) = -g^2 C_f \delta^{ab} \gamma_\mu \int_q \Pi_\perp^{\mu\nu}(q) G_A(q) G_q(p+q) \gamma_\nu. \tag{B.2}$$

Employing a split into the Dirac and mass components for the self-energy,

$$\Sigma_{\bar{q}q} = \Sigma_{\bar{q}q}^{(s)} + i\slashed{p}\,\Sigma_{\bar{q}q}^{(d)}, \tag{B.3}$$

one obtains

$$\Gamma_{\bar{q}q}^{(2)}(p) = i\slashed{p}\left(1 - \Sigma_{\bar{q}q}^{(d)}\right) + m - \Sigma_{\bar{q}q}^{(s)}(p), \tag{B.4}$$

By the fact that $\left(\Gamma_{\bar{q}q}^{(2)}\right)^{-1} = G_q$, we can now identify

$$
\begin{aligned}
Z_q(p) &= 1 - \Sigma_{\bar{q}q}^{(d)}(p), \\
M_q(p) &= \frac{1}{Z_q(p)}\left(m - \Sigma_{\bar{q}q}^{(s)}(p)\right).
\end{aligned} \tag{B.5}
$$

## B.1 Analytic calculation of the momentum integral

Using spectral representations for quark, (6), and gluon, (17), in the self-energy (B.2), one obtains

$$
\begin{aligned}
\Sigma_{\bar{q}q}(p) = &-g^2 C_f \delta^{ab} \gamma_\mu \int_{\lambda_A,\lambda_q} \lambda_A \rho_A(\lambda_A) \lambda_q \rho_q^{(s)}(\lambda_q) \int_q \Pi_\perp^{\mu\nu}(q) \gamma_\nu \frac{1}{q^2 + \lambda_A^2} \frac{1}{(p+q)^2 + \lambda_q^2} \\
&+ i g^2 C_f \delta^{ab} \gamma_\mu \int_{\lambda_A,\lambda_q} \lambda_A \rho_A(\lambda_A) \rho_q^{(d)}(\lambda_q) \int_q \Pi_\perp^{\mu\nu}(q) (\slashed{p}+\slashed{q}) \gamma_\nu \frac{1}{q^2 + \lambda_A^2} \frac{1}{(p+q)^2 + \lambda_q^2}.
\end{aligned} \tag{B.6}
$$

Using the split into Dirac and mass part of the self-energy (B.3), we identify

$$
\begin{aligned}
\Sigma_{\bar{q}q}^{(s)}(p) &= -g^2 C_f \delta^{ab} \int_{\lambda_A,\lambda_q} d\mu_\lambda^{(s)} I^{(s)}(p,\lambda_A,\lambda_q), \\
\slashed{p}\,\Sigma_{\bar{q}q}^{(d)}(p) &= g^2 C_f \delta^{ab} \int_{\lambda_A,\lambda_q} d\mu_\lambda^{(d)} I^{(d)}(p,\lambda_A,\lambda_q),
\end{aligned} \tag{B.7a}
$$

where we defined the spectral measures

$$
\begin{aligned}
d\mu_\lambda^{(d)} &:= \lambda_A \rho_A(\lambda_A) \rho^{(d)}(\lambda_q), \\
d\mu_\lambda^{(s)} &:= \lambda_A \rho_A(\lambda_A) \lambda_q \rho_q^{(s)}(\lambda_q),
\end{aligned} \tag{B.7b}
$$

and introduced the momentum integral functions

$$
\begin{aligned}
I^{(s)}(p,\lambda_A,\lambda_q) &= \gamma_\mu \int_q \Pi_\perp^{\mu\nu}(q) \gamma_\nu \frac{1}{q^2 + \lambda_A^2} \frac{1}{(p+q)^2 + \lambda_q^2}, \\
I^{(d)}(p,\lambda_A,\lambda_q) &= \gamma_\mu \int_q \Pi_\perp^{\mu\nu}(q) (\slashed{p}+\slashed{q}) \gamma_\nu \frac{1}{q^2 + \lambda_A^2} \frac{1}{(p+q)^2 + \lambda_q^2}.
\end{aligned} \tag{B.7c}
$$

Performing the Lorentz contractions and commutation of the Dirac structure yields

$$I^{(s)}(p, \lambda_A, \lambda_q) = (d-1) \int_q \frac{1}{q^2 + \lambda_A^2} \frac{1}{(p+q)^2 + \lambda_q^2} \,,$$

$$I^{(d)}(p, \lambda_A, \lambda_q) = \int_q \tau^{(d)} \frac{1}{q^2 + \lambda_A^2} \frac{1}{(p+q)^2 + \lambda_q^2} \,,$$

(B.8a)

with

$$\tau^{(d)} = \slashed{p}(3-d) + \slashed{q}\left(1 - d - 2\frac{p \cdot q}{q^2}\right).$$

(B.8b)

In order to get rid of the $1/q^2$-term in $\tau^{(d)}$, we apply partial fraction decomposition. This yields

$$I^{(d)}(p, \lambda_A, \lambda_q) = \int_q \left(\tau_1^{(d)} \frac{1}{q^2 + \lambda_A^2} + \tau_2^{(d)} \frac{1}{q^2}\right) \frac{1}{(p+q)^2 + \lambda_q^2} \,,$$

(B.9a)

where

$$\tau_1^{(d)} = \slashed{p}(3-d) + \slashed{q}\left(1 - d - 2\frac{p \cdot q}{\lambda_A^2}\right), \qquad \tau_2^{(d)} = -2\slashed{q}\frac{p \cdot q}{\lambda_A^2}.$$

(B.9b)

In a next step, we perform the Feynman trick on each product of two propagator kernels,

$$\frac{1}{q^2 + \eta_1^2} \frac{1}{(p+q)^2 + \eta_2^2} = \int_x \frac{1}{(q^2 + \Delta(\eta_1, \eta_2, p))^2} \,,$$

$$\Delta(\eta_1, \eta_2, p) = x\eta_1 + (1-x)\eta_2 + x(1-x)p^2 \,,$$

(B.10)

where $\int_x := \int_0^1 dx$. Performing the Feynman trick (B.10) necessitates shifting the loop momentum as $q \to q - xp$. This also shifts the tensor structure functions $\tau$. Also dropping off powers of loop momentum $q$, which vanish under the integral, and symmetrizing $\slashed{q}(p \cdot q) = \slashed{p}q^2/d$, yields

$$\tau_1^{(d)} \to \slashed{p}\tilde{\tau}_1^{(d)}, \qquad \text{with} \qquad \tilde{\tau}_1 = \left[3 - d + \frac{2}{d}\frac{p^2}{\lambda_A^2} + x\left(d - 1 + 2x\frac{p^2}{\lambda_A^2}\right)\right],$$

$$\tau_2^{(d)} \to \slashed{p}\tilde{\tau}_2^{(d)}, \qquad \text{with} \qquad \tilde{\tau}_2^{(d)} = \left[-\frac{2}{d}\frac{q^2}{\lambda_A^2} - 2x\frac{p^2}{\lambda_A^2}\right],$$

(B.11)

such that eventually

$$I^{(s)}(p, \lambda_A, \lambda_q) = (d-1) \int_q \int_x \frac{1}{\left(q^2 + \Delta(\lambda_A, \lambda_q, p)\right)^2} \,,$$

$$I^{(d)}(p, \lambda_A, \lambda_q) = \slashed{p} \int_q \int_x \tilde{\tau}_1^{(d)} \frac{1}{\left(q^2 + \Delta(\lambda_A, \lambda_q, p)\right)^2} + \tilde{\tau}_2^{(d)} \frac{1}{\left(q^2 + \Delta(0, \lambda_q, p)\right)^2} \,.$$

(B.12)

The momentum integrals can now be evaluated using standard integration formulae. Re-ordering the resulting expression in powers of the Feynman parameter $x$ and taking the limit

$d \to 4 - 2\varepsilon$, we arrive at

$$I^{(s)}(p, \lambda_A, \lambda_q) = \frac{3}{(4\pi^2)}\left(\frac{1}{\varepsilon} + \log\frac{4\pi\mu^2}{e^{\gamma_E}} - \int_x \log\Delta(\lambda_A, \lambda_q, p)\right) + \mathcal{O}(\varepsilon),$$

$$I^{(d)}(p, \lambda_A, \lambda_q) = \frac{\slashed{p}}{(4\pi)^2}\left\{\left(\frac{1}{\varepsilon} + \log\frac{4\pi\mu^2}{e^{\gamma_E}}\right)\sum_{i=0}^{3}\frac{\alpha_i + \beta_i}{i+1}\right.$$
$$\left. - \int_x \sum_{i=0}^{3} x^i \left(\alpha_i \log\Delta(\lambda_A, \lambda_q, p) + \beta_i \log\Delta(0, \lambda_q, p)\right)\right\} + \mathcal{O}(\varepsilon),$$
(B.13)

with $\gamma_E$ the Euler-Mascheroni constant. The coefficients $\alpha_i$ and $\beta_i$ appearing in the Dirac part $I^{(d)}$ of (B.12) do not depend on $x$, and will be given down below. Since we work in Landau gauge, the one-loop anomalous momentum is expected to vanish, i.e.,

$$\sum_{i=0}^{3}\frac{\alpha_i + \beta_i}{i+1} = 0,$$
(B.14)

serving as a consistency check for our calculation. Indeed, we find them to satisfy the condition (B.14).

The Feynman parameter integrals can be solved analytically. Using (B.14) and dropping the $1/\varepsilon$ term as well as the $\mathcal{O}(\varepsilon)$ contribution, we obtain the final result,

$$I^{(s)}(p, \lambda_A, \lambda_q) = \frac{3}{(4\pi^2)}\left(\log\frac{4\pi\mu^2}{e^{\gamma_E}} - f_0\right), \qquad I^{(d)}(p, \lambda_A, \lambda_q) = -\frac{\slashed{p}}{(4\pi)^2}\left(\alpha_i f_i + \beta_i g_i\right), \quad \text{(B.15)}$$

with $i = 0, ..., 3$. Using $C_f(SU(3)) = 4/3$ and $g^2 = 4\pi\alpha_s$, we ultimately arrive at

$$\Sigma_{\bar{q}q}^{(s)}(p) = -\frac{\alpha_s}{4\pi}\frac{4}{3}\delta^{ab}3\int d\mu_\lambda^{(s)}\left(\log\frac{4\pi\mu^2}{e^{\gamma_E}} - f_0\right),$$
$$\Sigma_{\bar{q}q}^{(d)}(p) = -\frac{\alpha_s}{4\pi}\frac{4}{3}\delta^{ab}\int d\mu_\lambda^{(d)}\left(\alpha_i f_i + \beta_i g_i\right).$$
(B.16)

The analytic result agrees quantitatively with that of [94]. Performing above calculation for arbitrary values of the gauge fixing parameter $\xi$, we reproduce the well-known one-loop perturbation theory results for wave function renormalisation $Z_\psi = 1 - \alpha_s C_f \xi$.

The coefficients $\alpha_i, \beta_i$ implicitly defined in (B.13) read

$$\alpha_0 = -2, \qquad \alpha_1 = -\frac{p_0^2 - 4\lambda_A^2 + \lambda_q^2}{\lambda_A^2}, \qquad \alpha_2 = \frac{3p_0^2}{\lambda_A^2},$$

$$\beta_0 = 0, \qquad \beta_1 = \frac{p_0^2 + \lambda_q^2}{\lambda_A^2}, \qquad \beta_2 = -\frac{3p_0^2}{\lambda_A^2}.$$
(B.17)

The functions $f_i$ and $g_i$ are defined as in Appendix A of [3] with $\lambda_1 = \lambda_A$, $\lambda_2 = \lambda_q$.

## B.2 Spectral renormalisation

The quark self-energy diagram is linearly divergent. While the momentum integrals are finite due to dimensional regularisation, the spectral integrals are not. This is due to the fact that the dimensional limit $\varepsilon \to 0$ is taken before performing the spectral integrals. This is necessary

however in order to solve the spectral integrals in a fully numerical fashion. In consequence, the spectral integrals require regularisation. To that end, we apply spectral BPHZ renormalisation. This scheme renormalises the spectral integral by subtracting a Taylor expansion of the spectral integrand such that the integral converges. For details on the procedure, we refer to [1]. Below, we only provide the renormalised expressions corresponding to the renormalisation conditions (15).

Since the superficial degree of divergence of the quark self-energy is one, it is sufficient to subtract the 0th order Taylor expansion. The renormalised DSE reads

$$\Gamma_{\bar{q}q}^{(2)}(p) = \mathrm{i}\not{p} + m - \left(\Sigma_{\bar{q}q}(p) - \Sigma_{\bar{q}q}(\mu)\right).\tag{B.18}$$

### B.3 Evaluation at real frequencies

In order to obtain the real-time expression for the quark self-energy diagram, in all coefficients $\alpha_i, \beta_i$ as well as in the functions $f_i$ and $g_i$ the substitution $p_0 \to -\mathrm{i}(\omega + \mathrm{i}0^+)$ is performed. The limit $0^+$ is taken analytically. It is crucial to stay on the correct side of the branch cut. A cross-check if this is the case is can be done by comparing the analytic limit to the numeric expression using a small, finite (positive) value for $0^+$.

### B.4 Iterative procedure

The spectral DSE is solved using a power iteration. The RHS of the DSE (11) is evaluated for a given quark spectral function $\rho_q^{(i)}$, where the superscript $^{(i)}$ now relates to the iteration number and not the component of the quark spectral function. We then calculate the spectral function of the next iteration $\rho_q^{(i+1)}$ via (7). This procedure is iterated until convergence of the pole position as well as the spectral tail by eyesight is reached. The iteration is initialised with quark spectral function corresponding to the classical propagator. In terms of vector and scalar component, they read

$$\rho^{(d/s)}(\omega) = \frac{\pi}{2}\delta(\omega - m).\tag{B.19}$$

Note that we omitted the $\omega < 0$ contributions in (B.19). They can be obtained via the symmetry relations (5).

## C Gauge parameter dependence of the dressing function

The infrared asymptotics of our result for dressing function $1/Z_q$, see Figure 4, differs qualitatively from that in other works in Landau gauge. Our result approaches a value above one, while in, e.g., [70, 72, 74, 83, 84] it is found that the dressing function saturates below one in the deep IR. In contrast to former references though, we only consider a classical quark-gluon-vertex in order to facilitate the real-time calculation. It is known that at one-loop order in perturbation theory, the dressing function does not receive quantum corrections. In Figure 7 we demonstrate the gauge parameter dependence of the IR behaviour of the quark dressing function in one-loop perturbation theory using a small gluon mass of $m_A = 10$ MeV. Although we are not using a massive gluon in obtaining our DSE results Figure 4, our gluon propagator has a characteristic mass scale given by the peak position of its spectral function, cf. Figure 3. In conclusion, we attribute the IR behaviour of our quark dressing function to the combination of the classical quark-gluon-vertex with a gapped gluon propagator. Including other tensor structures as well as a momentum-dependent dressing functions in the quark-gluon-vertex, we expect the IR value of the dressing function to drop below one, as seen in [70].

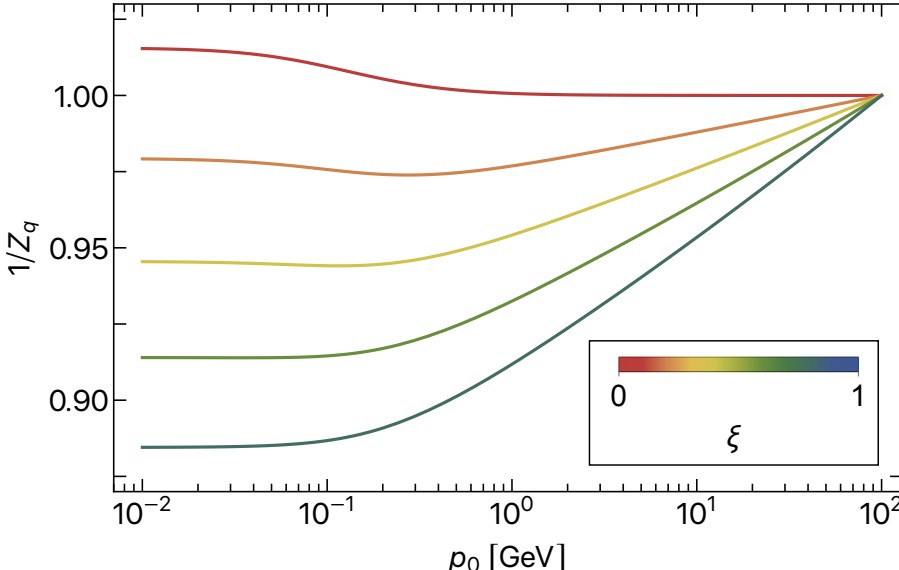

Figure 7: Gauge parameter dependence of the quark dressing function $1/Z$ at one-loop order in perturbation theory using a small gluon mass $m_A = 10$ MeV. A strong dependence of the IR asymptotic behaviour on the gauge parameter is observed. In Feynman gauge ($\xi = 1$), the dressing function is smaller than one in the deep IR. Similar to our non-perturbative Landau gauge results, in Landau gauge ($\xi = 0$) the dressing function rises above one.

## D  Numerical implementation

### D.1  Analytic structure during iterative solution

In this work, we solve the DSE iteratively via a power method, as described in Appendix B.4. There arises a subtlety when solving the DSE this way while employing the resonance-scattering split (A.14). Although the solution must exhibit a branch point singularity, this property usually does not hold while solving the equation. Staring from an initial guess for the spectral function with pole position $\omega_0$, the pole positions $\omega_i$ of subsequent iterations moves to the right, i.e., $\omega_i > \omega_0$. Thus, $\omega_i$ will always lie within the support of the mass functions imaginary part, such that instead of a branch point singularity, a very sharp peak appears. Once converged however, the pole position is no longer moving and directly lies on the branch point, and the branch point singularity appears. Technically, this property could be enforced by renormalisation during the iterations, i.e., we could use on-shell renormalisation to fix the pole position to always lie on the branch point. Since, while keeping the coupling constant fixed, this would modify the scales of our system which are already fixed by the gluon propagator input, we refrain from doing so.

### D.2  Determination of residues

Formally, the residues of the delta pole contributions in the resonance-scattering split (10) are related to the quark propagator by (A.15). It turned out to be numerically more stable to determine them by a fit to the Euclidean propagator data instead. For that, a momentum scale $p_{\text{fit}}$ needs to be chosen. For the residue of the scalar part, we chose $p_{\text{fit}} = \mu = 30$ GeV. This ensured that also the propagator obtained from the spectral representation respects the renormalisation condition for the mass function. This is in fact important for numerical stability. The mass function becomes very small in the UV. A lack of numerical precision can

violate positivity of the Euclidean mass function (obtained through the spectral representation) in the UV. These negative parts introduce unstable directions in the iteration which must be avoided in order to avoid a solution.

For the vector part, we specified $p_{\text{fit}} = 1$ GeV.

### D.3 Spectral integration domain

In the resonance-scattering split (10), an onset for scattering tail needs to chosen such that pole and tail contribution are well satisfied. This needs to be done such that the pole contribution has decayed sufficiently without neglecting relevant scattering contributions. The onset scale is chosen such that the resonance-scattering split is reproduces the Euclidean propagator as good as possible, see Appendix A.1. We use $\omega_{\text{onset}} = m_{\text{pole}} + 0.1$ GeV, where $m_{\text{pole}}$ is the pole position of the respective resonance contribution.

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
