# Peer review of "On the quark spectral function in QCD"

_SciPost Physics, doi:SciPost Phys. 15, 149 (2023)_

## Round 1 · Referee Report · Anonymous (Referee 2) · 2023-7-27

Report

I thank the authors for their response and for taking some of my suggestions into account. However, the following points still need to be addressed before I can recommend publication.

1) I still believe that the quality of the paper would strongly benefit from adding an explicit (analytic and numerical) example to illustrate condition (9) based on one of the standard propagators from the literature which has a well-known analytic structure, for example as shown in Fig. 3.9 (right) of https://arxiv.org/abs/1606.09602 or discussed by Alkofer-Watson-Weigel (Phys. Rev. D 65, 094026). If the discussion in Appendix A covers such a case, this should be mentioned and substantiated by explicitly using the corresponding numerical parameters.

3) If the authors insist on using this graphical notation, it should at least be justified in the paper and clearly mentioned that this deviates from the wide-spread convention in the literature, in order to avoid confusion of the reader.

4) Concerning my request to mention (10) already in the abstract, the authors respond that "Since we consider this to be a calculational, technical detail, we decided against mentioning this in the abstract."

However, in the manuscript, below Eq. (10), it is written that "Equation (10) represents a central approximation of this work."

If it is so central, it must be mentioned in the abstract.

  • validity: -
  • significance: -
  • originality: -
  • clarity: -
  • formatting: -
  • grammar: -

Author:  Jan Horak  on 2023-07-31  [id 3853]

(in reply to Report 1 on 2023-07-27)

We thank the reviewer for the additional suggestions. Below, we reply to their comments as numbered in the report.

1) To accommodate the referee’s request for an explicit example of the condition (9), below (9), we added a summary of the discussion of Appendix A.

We hope that the readers will be able to convince themselves that the roots of

                 x^2 - (z ± i * y)^2 = 0           for         z ,y real,

lie on the first/second Riemann sheet for the +/- case, and that this example does not require visualisation.

2 ) To avoid confusion, we added a comment that in contrast to other diagrammatic notations, in our notation full propagators do not carry blobs and the inverse bare propagator is simply given by a dot with two legs.

In our opinion, the only requirements for notation are clarity and consistency–both of which are fulfilled in our case. Beyond that, notation should be completely up to the authors.

3) We added a subsentence mentioning the approximation (10) in the abstract.

---

## Round 1 · List of Changes

- added an explanation of \omega_0 below eq. (9)
- we added a comment mentioning the approximation (22) in the introduction

---

## Round 2 · Referee Report · Anonymous (Referee 2) · 2023-8-9

Strengths

To the best of my understanding this is a relevant discussion of the analytic properties of the quark and gluon propagator. The authors have, in my opinion, taken the criticisms of the previous reviewing rounds on board, thereby improving the quality of the manuscript to a level where I am happy to support its publication.

Report

I think the criteria for publication are met - I support its speedy processing.
As the editor-in-charge found it difficult to recruit reviewers (an ongoing issue across the board and across journals that needs addressing), I was asked to look at the submission, without being an expert in the field. I therefore read the paper and reviewed its refereeing and submission history and found that the authors have by far and large met the criticisms of the better qualified referee, to a level where publication can be supported.

---

## Round 2 · Referee Report · Anonymous (Referee 1) · 2023-8-14

Report

The authors have taken care of all my remarks and suggestions. I now recommend publication.

---

## Round 2 · List of Changes

• added a summary of the discussion of Appendix A below (9)
  • added a comment that in contrast to other diagrammatic notations, in our notation full propagators do not carry blobs and the inverse bare propagator is simply given by a dot with two legs below Fig. 2
  • added a subsentence mentioning the approximation (10) in the abstract

---

## Editorial Decision

published